# PLAY2PROMPT: ZERO-SHOT TOOL INSTRUCTION OPTIMIZATION FOR LLM AGENTS VIA TOOL PLAY

## ABSTRACT

Large language models (LLMs) are increasingly integrated with external tools to complete user requests. Many real-world applications require LLMs to use specialized tools in a zero-shot setting. To achieve this, current methods primarily rely on prompting LLMs with tool-specific information, yet tool documentation is often underspecified or noisy, limiting effectiveness. Manual improvements are inefficient and impractical, as they require domain expertise to rewrite documentation and test on carefully curated held-out datasets to evaluate performance gains. Automatic prompt engineering techniques are not applicable either, because they require labeled examples, which is unavailable in the zero-shot setting. In this work, we introduce PLAY2PROMPT, an automated framework that iteratively refines tool documentation and generates usage examples. PLAY2PROMPT enables LLMs to explore tool input-output behaviors, allowing us to effectively search the space of possible tool descriptions and examples. The generated examples not only guide LLM inference but also serve as validation data to ensure more effective tool use. Extensive experiments on real-world tasks demonstrate significant improvements in zero-shot tool performance across both open- and closed-source models.

## 1 INTRODUCTION

Recently, there has been growing research interest in enhancing large language models (LLMs) by integrating external tools with specialized capabilities. This augmentation allows for automatic planning and execution of tool usage, thereby enabling LLMs to solve complex tasks with greater accuracy and produce responses that are more aligned with human preferences (Mialon et al., 2023; Qin et al., 2024a). For instance, open-source models have been fine-tuned on manually curated or synthetically generated function-calling data to improve performance in specific reasoning and question-answering tasks (Schick et al., 2023; Yang et al., 2023). Additionally, both open-source base models and closed-source black-box models are being trained to invoke a limited set of built-in tools, such as mathematical calculators or search engines. However, while these tools address general use cases, they often prove insufficient for real-world complicated tasks that require domain-specific functionalities. Therefore, it is essential to develop methods that enable these tool-use frameworks to dynamically learn how to use user-defined tools.

Training models to specialize in new tools necessitates extensive fine-tuning data and significant computational resources, rendering this approach impractical for large-scale applications. A more viable alternative involves augmenting the set of built-in tools by supplementing user-defined tools at inference time in a zero- or few-shot manner via prompting (Lu et al., 2023; Shen et al., 2023). This method capitalizes on the zero-shot tool-calling capabilities of current LLMs, which have been tuned with tool-use instructions to facilitate this plug-and-play functionality.

One typical such paradigm is ReAct (Yao et al., 2023), wherein the model plans and selects appropriate tools to accomplish the given task, interleaving reasoning steps with tool retrieval, tool call predictions, and executions. In the ReAct paradigm, the success of learning to use new tools, particularly in zero-shot scenarios, hinges on *comprehensive tool documentation and demonstrations* (Hsieh et al., 2023; Patil et al., 2023). Such documentation generally consists of tool descriptions, parameter specifications, output formats, and other related meta-information, which provides critical information for equipping the LLM with necessary information to utilize the tools correctly.

Figure 1: The PLAY2PROMPT framework: Beam search iteratively searches demonstrations, incorporating tool play into the exploration and self-reflection process (Left). After demonstrations are optimized, beam search is once again applied to optimize descriptions by evaluating on the demonstrations as test set, and incorporating tool use outputs/errors (Right).

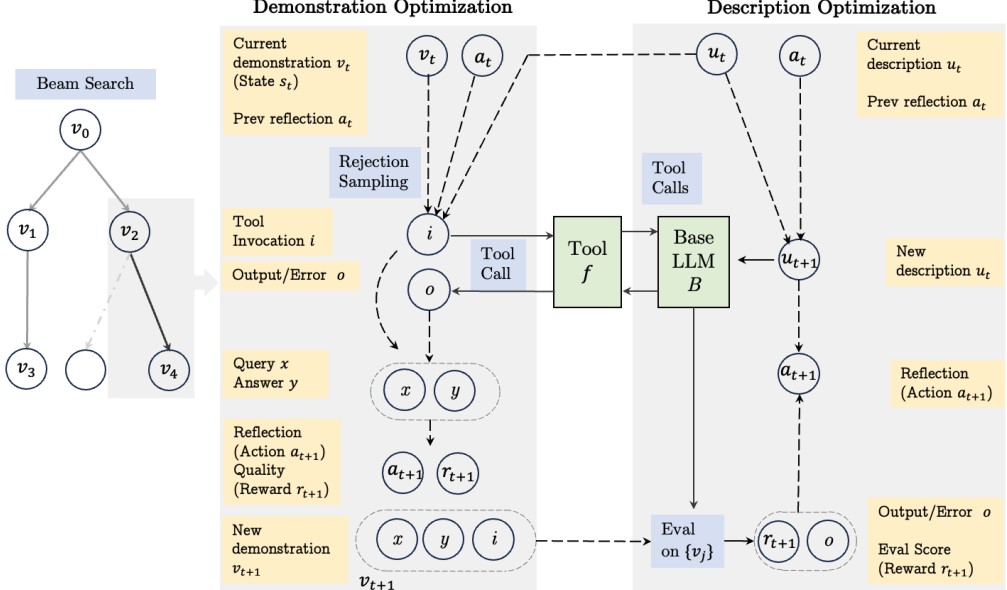

However, in numerous practical cases, users fail to provide adequate documentation or exemplar demonstrations to the model, nor do they invest in crafting improved documentation tailored for LLM utilization. This lack of information can lead to failures in tool usage, such as syntax errors in both zero-shot and fine-tuned models (Zhang et al., 2023a), hallucinations due to lack of proper tool documentation (Hsieh et al., 2023), and diminished performance resulting from insufficient demonstrations (Xu et al., 2023). Manually enhancing tool documentation and creating example demonstrations is laborious and inefficient, which is further compounded by the need for labeled testing data for each tool to assess effectiveness. When attempting to scale up to larger API databases or online code repositories, these challenges become even more pronounced.

While automatic prompt engineering techniques have been shown to outperform manual optimization (Wang et al., 2024), they prove inapplicable in this context due to their reliance on labeled examples for testing—resources that are inherently unavailable in zero-shot settings (Wu et al., 2024). Existing methods for revising tool documentation typically involve directly prompting LLMs to optimize tool descriptions (Yuan et al., 2024), lacking the capacity to evaluate whether the rewritten documentation enhances the LLM's tool use performance, instead depending heavily on meticulously crafted meta-prompts and oracle example demonstrations supplied within the meta-prompts.

To address these challenges and facilitate general zero-shot tool utilization, we introduce PLAY2PROMPT, an automated framework, which iteratively refines tool documentation and generates example tool usage demonstrations, as illustrated in figure 1. PLAY2PROMPT does not rely on any external tool use examples. Instead, drawing inspiration from human trial-and-error methodologies, it prompts an LLM agent to "play" with the new tools to explore their functionalities and usage, based on which the tool use examples and refined tool descriptions are generated.

During each generation process, PLAY2PROMPT executes multiple trial-and-error iterations, leveraging both successful and erroneous tool-use instances to guide the search trajectory. We employ self-reflection (Madaan et al., 2023; Pryzant et al., 2023; Shinn et al., 2023) to generate error feedback, thereby directing the search algorithm towards progressively improved outputs. Crucially, we iteratively refine not only the tool descriptions but also generate example demonstrations. The generated examples function as a validation set, enabling LLMs to interact with and evaluate tool usage, which subsequently guides the further enhancement of tool descriptions. Throughout this iterative process, PLAY2PROMPT systematically expands the search space in a tree structure, pri-

oritizing paths with higher self-reflection or evaluation scores. PLAY2PROMPT operates entirely in a zero-shot manner and is inherently task-agnostic, making it a practical and scalable solution for enhancing LLM tool utilization without necessitating additional labeled data or manual intervention.

We demonstrate the effectiveness of PLAY2PROMPT by applying it to real-world scenarios. On the StableToolBench benchmark (Guo et al., 2024), our approach consistently surpasses baseline methods for both open-source LLaMA models (Dubey et al., 2024) and closed-source OpenAI GPT models (Achiam et al., 2023). Extensive experiments and analyses further underscore the efficacy of our approach.

Our contributions can be summarized as follows:

- We introduce PLAY2PROMPT, a novel automated framework that iteratively refines tool documentation and generates usage examples, empowering LLMs to utilize tools more effectively in zero-shot settings without the need for labeled data.
- PLAY2PROMPT integrates a search-based trial-and-error process augmented with self-reflection, enabling LLMs to interact with tools, explore their functionalities, and iteratively refine both tool descriptions and demonstrations, thereby significantly enhancing performance.
- PLAY2PROMPT is entirely zero-shot, inherently scalable, and task-agnostic, making it broadly applicable across a wide range of tools and domains, and practical for enhancing LLM tool use at scale without additional manual effort.

## 2 METHODOLOGY

Tool documentation typically includes tool descriptions, parameter specifications, output formats, and other related meta-information. In our framework, we define a tool as $f = (u, I, g)$, where $u$ denotes the tool description, $I$ represents tool-related meta-information (such as parameter specifications and other relevant details), and $g$ is the corresponding executable function call. Example tool usage demonstrations often vary from simple question-answer pairs to comprehensive reasoning chains. We focus on demonstrations that illustrate when and how the tool can be used; thus, we define an example demonstration as $v = (x, y, i)$, comprising a question $x$, an answer $y$, and a tool invocation $i$ that specifies the tool call parameters. When utilizing tools during inference in the ReAct paradigm, LLMs typically interleave reasoning steps with tool invocation and execution in a chain-of-thought manner. Given an input user query $x$, a base LLM $\mathcal{B}$, a set of available tools $F = \{f_j\}_{j=1}^{K}$, and a set of example demonstrations $V_F = \{v_j\}_{j=1}^{M}$ for the tools $F$, we denote the entire ReAct chain as $\mathcal{B}(x; F; V_F)$.

**Problem Formulation** Consider a set of testing samples $D_{\text{test}} = \{x_j, y_j, F_j\}_{j=1}^{N}$. The evaluation of the tool-using ability of the base model $\mathcal{B}$ is given by $\mathbb{E}_{D_{\text{test}}}[\texttt{Score}(\mathcal{B}(x_j; F_j; V_{F_j}), y_j)]$, where $\texttt{Score}$ is a scoring function assessing the alignment between the model's output and the ground truth $y_j$. The primary objective is to maximize this evaluation score by improving tool descriptions and generating effective example demonstrations. In a zero-shot scenario, we lack access to labeled testing samples and the specific tool sets $F_j$ for each test sample, rendering direct optimization of the objective infeasible. Therefore, we require a proxy testing set $D_{\text{proxy}}$ and a corresponding tool set $F_{\text{proxy}}$. If a small validation set is available, as in automatic prompt optimization settings, it can serve as the proxy testing set and tool set. In our setting, we assume access to a new tool $t \in \cup_j F_j$ and treat each incoming tool $f$ independently by setting $F_{\text{proxy}} = \{f\}$. Assuming a fixed number of demonstrations $M$, the optimization objective for the tool description $u$ and example demonstrations $\{v_j\}_{j=1}^{M}$ becomes $u^*, \{v_j^*\}_{j=1}^{M} = \arg\max_{u \in \mathcal{U}, v_j \in \mathcal{V}} \texttt{Score}(\mathcal{B}(x, \{(u, I, g)\}, \{v_j\}_{j=1}^{M}), y)$, where $\mathcal{U}$ and $\mathcal{V}$ represent the sample spaces for tool descriptions and example demonstrations, respectively. Given the vastness of these spaces, it is essential to design an algorithm that can search through them efficiently and effectively. To this end, we propose a framework that iteratively generates the demonstrations $V_F$, assigns $D_{\text{proxy}} = \{x_j, y_j, F\} \, \forall (x_j, y_j, i_j) \in V_F$, and refines the descriptions $u$. In the following sections, we introduce PLAY2PROMPT and detail its two optimization tasks.

### 2.1 PLAY2PROMPT

The primary goal of PLAY2PROMPT is to integrate knowledge gained from tool interactions into tool usage descriptions and example demonstrations while ensuring an efficient exploration of the

large search space. Inspired by automatic prompt optimization methodologies (Wang et al., 2024), we devise the optimization process as a search framework where each state $s$ represents an iteration of the variable being optimized—either a demonstration ($s = v$) or a tool description ($s = u$). Each action $a$ corresponds to a modification applied to the current state.

To navigate the search space toward higher-quality regions, we generate actions based on inputs and outputs obtained from tool interactions. Feedback from tool executions—including successful outputs and usage errors—guides further revisions, ensuring that the updated state helps the base model $\mathcal{B}$ avoid previous mistakes. This iterative refinement process is influenced by prior work on self-reflection capabilities in LLMs (Shinn et al., 2023; Pryzant et al., 2023).

Specifically, given a state $s_t$, an action $a_t$ is generated by sampling from an optimization model $\mathcal{M}$, conditioned on the input-output information obtained from tool interactions. Applying the action $a_t$ to the state $s_t$—also performed via sampling from $\mathcal{M}$—yields the next state $s_{t+1}$. A scoring function evaluates the quality of each state, assigning a score $r_t$ that reflects the effectiveness of the current tool description or demonstration. This formulation allows for the integration of search algorithms to efficiently traverse the search space. In our work, we employ beam search to identify high-scoring states, treating each state as a node in a tree and exploring branches for potential improvements. The sampling strategies for generating $s_{t+1}$ and $a_{t+1}$, as well as the reward definitions, differ between the two optimization tasks. These details are elaborated in sections 2.2 and 2.3.

The two optimization tasks are inherently interdependent: refining tool descriptions requires evaluation examples to calculate a score $r$, while generating high-quality example demonstrations depends on detailed and accurate tool descriptions. By iteratively alternating between generating demonstrations and refining descriptions, each step informs the other: improved demonstrations highlight areas where the tool description may be lacking, while refined descriptions enable the generation of more accurate and effective demonstrations. This synergy allows PLAY2PROMPT to progressively enhance both components, ultimately improving the base model's tool-using capabilities.

## 2.2 TOOL EXAMPLE DEMONSTRATION OPTIMIZATION

The objective of this task is to generate example tool usage demonstrations for a given tool $f$, utilizing its initial tool description $u$, meta-information $I$, and function $g$, with the assistance of an optimization model $\mathcal{M}$. Directly sampling query-answer pairs $(x_{t+1}, y_{t+1}) \sim p_{\mathcal{M}}(x, y \mid x_t, y_t, u, I)$ poses significant challenges, because generating high-quality queries is difficult given only potentially incomplete or noisy tool descriptions and parameter information, especially without any other query-answer demonstrations in zero-shot settings. Additionally, constraining the scope of the generated query to only the specific tool is challenging; generated queries might require the use of other tools, thus expanding the search space beyond manageable limits.

To overcome these issues, we adopt an alternative approach by first sampling the tool invocation $i$ and then generate the corresponding query $x$ and answer $y$. This method leverages the fact that the search space for tool input parameters is considerably smaller and more constrained, especially when informed by the parameter specifications in $I$. By sampling a tool invocation $i$ first, we can execute it using the function $g$ to obtain the output $o = g(i)$. This step not only validates the tool call but also provides concrete input-output examples of the tool's functionality. Generating the query $x$ conditioned on a valid tool call $i$ and its output $o$ benefits from this additional information, resulting in more relevant and focused demonstrations. The optimization model $\mathcal{M}$ thus gains substantial insight from interacting with the tool, effectively narrowing the search space.

Our sampling strategy consists of two stages. In the first stage, we perform rejection sampling of the tool invocation. We sample a candidate tool invocation $i \sim p_{\mathcal{M}}(\cdot)$ using the optimization model $\mathcal{M}$, execute the tool function to obtain $o = g(i)$, and verify whether $o$ is a valid output given the meta-information $I$ and description $u$ with $\mathcal{M}$. If the invocation is invalid, we reject it and repeat the sampling process until a valid tool invocation is obtained.

In the second stage, once a valid tool invocation $i$ is secured, we proceed to generate the corresponding user query $x$ and answer $y$. We sample the query $x \sim p_{\mathcal{M}}(\cdot|i)$ and then the answer $y \sim p_{\mathcal{M}}(\cdot|x, i)$. To refine these samples and enhance their quality, we perform a rollout of $N_{\text{refine}}$ steps, with self-reflection acting as the policy guiding the refinement process. During each rollout step, the model evaluates the alignment of the query, answer, and tool output, and generates self-

---

**Algorithm 1** DEMONSTRATIONSTATETRANSITION

---

**Input:** $s_t = v_t = (x_t, y_t, i_t)$: demonstration, $u$: description, $a_t$: reflection
**Output:** $s_{t+1} = v_{t+1} = (x_{t+1}, y_{t+1}, i_{t+1})$: demonstration, $r_{t+1}$: score, $a_{t+1}$: reflection

1: $c \leftarrow$ false
2: **while** $\neg c$ **do**                                          ▷ Rejection Sampling of tool invocation $i$
3:     $i_{t+1} \sim p_\mathcal{M}(i|i_t, c, u, I, a_t, m_1)$       ▷ Sample candidate tool invocation, incorporating reflection $a_t$
4:     $o_{t+1} \leftarrow g(i_{t+1})$                                   ▷ Execute tool function
5:     $c \sim p_\mathcal{M}(c|i_{t+1}, o_{t+1}, u, I, m_2)$         ▷ Verify validity of tool call
6: **end while**
7: **for** $n \leftarrow 1$ to $N_{\text{refine}}$ **do**                 ▷ Rollout for with self-reflection policy
8:     $x_{t+1} \sim p_\mathcal{M}(x|i_{t+1}, o_{t+1}, u, I, m_3)$     ▷ Sample user query $x$
9:     $y_{t+1} \sim p_\mathcal{M}(y|i_{t+1}, o_{t+1}, x_{t+1}, u, I, m_4)$   ▷ Sample corresponding answer $y$
10:     $r_{t+1} \sim p_\mathcal{M}(r|y_{t+1}, i_{t+1}, o_{t+1}, x_{t+1}, u, I, m_5)$   ▷ Evaluate demonstration quality
11:     $a_{t+1} \sim p_\mathcal{M}(a|r_{t+1}, y_{t+1}, i_{t+1}, o_{t+1}, x_{t+1}, u, I, m_6)$   ▷ Generate self-reflection action
12: **end for**

---

reflection actions to iteratively improve them. For scoring, we compute a reward $r_{t+1}$ by querying the model $\mathcal{M}$, conditioned on the sampled $i$, $x$, and $y$. This score reflects the quality and coherence of the demonstration. Finally, we generate a self-reflection action $a_{t+1}$ to guide further optimization. The detailed procedure is outlined in Algorithm 1, where each $m_i$ is a meta-prompt.

### 2.3 TOOL DESCRIPTION OPTIMIZATION

Optimizing tool descriptions requires a strategy that effectively incorporates feedback from the base model's tool usage, particularly when errors occur. During a state transition, we sample a new tool description $u_{t+1} \sim p_\mathcal{M}(u|u_t)$ using the optimization model $\mathcal{M}$. To evaluate the effectiveness of this new description, we utilize the previously generated example demonstrations $V = \{(x_j, y_j, i_j)\}_{j=1}^M$. We calculate the score $r_{t+1}$ by testing the base model $\mathcal{B}$ in the ReAct framework on the demonstration set $V$ using $u_{t+1}$, the new tool description: $r_{t+1} = \mathbb{E}_j[\text{Score}(\mathcal{B}(x_j, \{(u_{t+1}, I, g)\}, \{\}), y_j)]$.

A critical aspect of this optimization task is incorporating tool-use information derived from the base model's interactions with the tool. When the base model $\mathcal{B}$ uses the tool incorrectly—resulting in errors such as invalid parameter usage, incorrect function calls, or misinterpretation of the tool's purpose—the errors provide valuable feedback. These errors, along with the tool outputs, are collected during the ReAct chains. We condition the generation of the self-reflection action $a_{t+1}$ on this collected information.

By analyzing the errors encountered, the optimization model $\mathcal{M}$ can identify deficiencies or ambiguities in the current tool description $u_{t+1}$ that may have contributed to the incorrect usage. The self-reflection action $a_{t+1}$ then suggests specific modifications to the tool description aimed at mitigating these issues. For instance, if the base model frequently misuses a parameter due to unclear specifications, the self-reflection process may recommend clarifying that parameter's description or providing examples of correct usage. Similarly, if the base model misunderstands the overall functionality of the tool, the reflection may suggest rephrasing the tool description to be more explicit. By iteratively refining the tool description in response to observed errors, we enhance the base model's ability to use the tool correctly in future interactions, reducing the likelihood of repeated mistakes. This iterative refinement process not only improves the clarity and usefulness of the tool description but also contributes to more effective and efficient tool usage by the base model. The detailed procedure for tool description optimization is presented in Algorithm 2.

## 3 EXPERIMENTS

### 3.1 EXPERIMENTAL SETUP

**Task**   To assess the effectiveness of tool instruction optimization in real-world applications, we evaluate on StableToolBench (Guo et al., 2024), a benchmark containing diverse user requests across

---

**Algorithm 2** DESCRIPTIONSTATETRANSITION

---

**Input:** $s_t = u_t$: description, $V = \{(x_j, y_j, i_j)\}_{j=1}^M$: demonstration set, $a_t$: reflection

**Output:** : $s_{t+1} = u_{t+1}$: description, $r_{t+1}$: score, $a_{t+1}$: reflection

1: $u_{t+1} \sim p_{\mathcal{M}}(u|u_t, a_t, I, m_7)$         ▷ Sample description $u_{t+1}$ from $\mathcal{M}$, applying reflection $a_t$

2: $o_j, e_j \leftarrow \mathcal{B}(x_j, \{(u_{t+1}, I, g)\}, \{\}) \, \forall j$ ▷ Gather I/O & errors $e_j$ from running $\mathcal{B}$ on demo set $V$ with $u_{t+1}$

3: $r_{t+1} \leftarrow \mathbb{E}_j[\text{Score}(o_j, y_j)]$               ▷ Evaluation score on $V$

4: $a_{t+1} \sim p_{\mathcal{M}}(a|u_{t+1}, r_{t+1}, I, \{x_j, o_j, e_j\}, m_8)$      ▷ Self-reflection action

---

a large set of publicly available REST APIs from the RapidAPI Hub. StableToolBench improves upon the commonly used ToolBench (Qin et al., 2024b) by addressing the instability of RapidAPIs in the original version. If API access is unavailable, the benchmark employs a fallback system that uses caching and an API simulator. StableToolBench includes 16,464 APIs spanning 49 categories. Our experiments cover all six subsets of the benchmark, which include single-tool (I1) and multi-tool (I2-same category and I3-different category) test cases. In this context, an API service represents a tool that contains multiple sub-tools, with each sub-tool corresponding to $f$ in our definition. Thus, I1 test queries often require multiple sub-tool calls within a tool, making them not strictly "single-tool." Although the original subsets evaluate different types of generalizability based on tool overlap with training data, our zero-shot setting does not rely on any training data, rendering these differences less significant.

**Inference and Evaluation** We adhere to the inference setting used in the original benchmark, where a set of tools is provided for the base LLM $\mathcal{B}$ to select from to answer user queries. ReAct serves as our baseline performance method, for which we run on the testing data using the ReAct prompts provided in dataset, with the original tool descriptions and no example demonstrations, as we operate in zero-shot setting. For PLAY2PROMPT, we run ReAct again but with the optimized descriptions and demonstrations. To test different base models $\mathcal{B}$, we evaluate with Meta LLaMA models and OpenAI GPT models, both of which are trained with tool use instructions and have zero-shot tool-calling capabilities. Specifically, we tested `llama-3-8b-instruct` and `llama-3-70b-instruct` for LLaMA, while `gpt-3.5-turbo-1106` was used for GPT experiments. Since ReAct outputs are free-form, an evaluation LLM determines whether a response adequately answers a user query. Following the original benchmark's evaluation pipeline, we reuse the provided prompts and employ solvable pass rate as our evaluation metric, which measures the percentage of queries deemed solvable by the evaluation LLM. We use `llama-3.1-70b-instruct` as the evaluation LLM, as previous work (Guo et al., 2024) reported potential evaluation instabilities with weaker models, which we do not observe with this evaluation LLM. Additional details on inference and evaluation are provided in appendix A.

**Optimization Details** For PLAY2PROMPT, we follow the proposed algorithms, first running beam search to optimize example demonstrations. $N_{\text{refine}}$ is set to 5 for the demonstration optimization procedure. We set a depth limit of 5, beam width of 3 and conduct 3 explorations per node. The top 3 examples are generated and selected for each tool, which are then passed to the description optimization phase. Beam search is again applied with the same settings to select the best tool description. `llama-3-8b-instruct` is used for $\mathcal{M}$.

### 3.2 RESULTS AND ANALYSES

**Main Results on StableToolbench** The top and bottom rows for each base model in table 1 show the solvable pass rates of running ReAct with demonstrations and descriptions generated by PLAY2PROMPT, compared to the baseline of ReAct with original descriptions and no demonstrations. Across all 6 subsets, we see that PLAY2PROMPT outperforms on all base models, observing 3-6% absolute gains (6-9% relative gains) on average across all base models. For the smaller model LLaMA-3-8B, gains mainly come from I1-Cat, I1-Tool, and I3-Inst subsets, while the larger GPT-3.5 and LLaMA-3-70B models maintain consistent gains across all subsets. It is noteworthy that LLaMA-3-70B achieves the highest performance out of all 3 models, for both baseline ReAct and PLAY2PROMPT, and especially performs well on I3-Inst, the most challenging subset, where the other models struggle with the most. PLAY2PROMPT essentially boosts performance of models up

Table 1: Results on StableToolBench. Scores indicate the solvable pass rate.

| Base Model | Method | I1-Inst | I1-Cat | I1-Tool | I2-Inst | I2-Cat | I3-Inst | Avg |
|---|---|---|---|---|---|---|---|---|
| LLaMA-3-8B | ReAct | 59.2 | 60.8 | 55.8 | 58.1 | 56.7 | 47.7 | 56.4 |
| | PLAY2PROMPT - Desc | 59.9 | **65.9** | 57.7 | 58.0 | 56.7 | 50.8 | 58.1 |
| | PLAY2PROMPT - Demo | 57.9 | 65.5 | 58.5 | 58.0 | 56.7 | **59.4** | 59.3 |
| | PLAY2PROMPT | **60.0** | 65.6 | **61.0** | 59.0 | 57.8 | 53.4 | **59.5** |
| LLaMA-3-70B | ReAct | 70.3 | 75.9 | 66.1 | 70.6 | 76.4 | 76.5 | 72.7 |
| | PLAY2PROMPT - Desc | 71.1 | 78.0 | 66.5 | 76.1 | 78.2 | 79.2 | 74.9 |
| | PLAY2PROMPT - Demo | **73.6** | 78.5 | 71.8 | 76.3 | **83.1** | 76.3 | 76.6 |
| | PLAY2PROMPT | **73.6** | 79.4 | 72.5 | 76.7 | 80.9 | 80.3 | **77.2** |
| GPT-3.5 | ReAct | 57.4 | 67.8 | 65.1 | 61.2 | 62.9 | 53.0 | 61.2 |
| | PLAY2PROMPT - Desc | 60.1 | 67.5 | 66.0 | 61.9 | **67.9** | 55.7 | 63.2 |
| | PLAY2PROMPT - Demo | **62.2** | 70.2 | 70.3 | **64.6** | 65.4 | 56.6 | 64.9 |
| | PLAY2PROMPT | 62.0 | **70.7** | **71.7** | **64.6** | **67.9** | **64.7** | **66.9** |

to the baseline performance of models that are much larger. This underscores the effectiveness of PLAY2PROMPT, operating entirely without supervision with only access to the newly given tool itself.

**Effects of Demonstrations vs Descriptions**   To study the effectiveness of PLAY2PROMPT, we break down how much the newly optimized demonstrations contribute to the performance gains compared to the optimized descriptions. We keep our optimization procedure the same, but during inference we add two other settings: one where we use optimized demonstrations along with original descriptions, and one where we do not use demonstrations but update the descriptions with optimized ones. The results can be found labeled as PLAY2PROMPT-Desc and PLAY2PROMPT-Demo for each model in table 1. Compared against each other, we observe that the optimized example demonstrations generally contribute more to performance than optimized descriptions do. However, we still observe several subsets where one does well while the other remains close to the baseline. This varies across different base models, and we do not observe a pattern of when to choose one over another. On the other hand, using both together consistently obtains the best performance, especially for the larger models, giving us higher assurance of performance gains. This suggest that information from the optimized descriptions and demonstrations may complement each other in assisting LLMs' tool use, validating our iterative approach.

**Ablation on Search Strategies**   To investigate the effect of search effectiveness in PLAY2PROMPT, we conduct an ablation study by comparing beam search to alternative search strategies. Specifically, we compare to Monte Carlo (MC) search with different depths, using the same transition and action strategy as in PLAY2PROMPT and only replace beam search with MC. MC in this context would be a single step of sampling a state and an action. In this ablation, we run MC with depth of 1 and 5. Additionally, we aim to quantify the effect of having $N_{\text{refine}}$ rollout steps with the self-reflection policy when sampling query-answer pairs, and compare against a strategy of not rolling out. We run the experiments on I3-Inst, which is generally the hardest subset due to the inclusion of tools across categories. The results are presented in table 2, where MC is shown to perform worse than beam search (PLAY2PROMPT), as it does not explore the search space well enough to reach better states. Rollout refinement helps, as does increasing the depth of search due to information gained from tool play and self-reflection actions.

**Analysis on Single-Tool Queries and Incomplete Descriptions**   We attempt to gain more insight into how the optimized demonstrations and descriptions enhance base model's tool use by focusing on certain user queries of interest. One such set contains queries that use only a single sub-tool, as it most closely matches to our optimization scenario, where we denote as the I1-Sub subset. Another interesting set are instances where a query's corresponding tool set contains at least one tool that lacks tool description, i.e., only has meta-information in the form of tool names and parameter structures. We denote this set as NoDesc. Evaluation results on these two subsets are shown in table 3. PLAY2PROMPT greatly enhances the performance of LLaMA-3-8B on both I1-Sub and NoDesc, and almost reaches the performance of the 70B model on I1-Sub. However, the 70B model

Table 2: Ablation on search strategies, ran on I3-Inst. LLaMA-3-8B is used for $\mathcal{B}$ in this experiment. MC denotes Monte Carlo search.

| Search strategy | I3-Inst |
|---|---|
| MC (depth= 1) | 48.1 |
| MC (depth= 5) | 49.9 |
| MC (depth= 5) & $N_{\text{refine}} = 5$ | 51.9 |
| Beam search (PLAY2PROMPT) | 53.4 |

Table 3: Solvable pass rate on instances only using a single subtool (I1-Sub) and instances whose toolset includes at least one tool without a tool description (NoDesc). Demo indicates performance evaluated on the generated demonstration set.

| Base Model | Method | I1-Sub | | NoDesc | |
|---|---|---|---|---|---|
| | | demo | test | demo | test |
| LLaMA-3-8B | ReAct | 85.0 | 59.7 | 85.0 | 57.5 |
| | ReAct+PLAY2PROMPT | 93.6 | 72.4 | 98.5 | 72.9 |
| LLaMA-3-70B | ReAct | 92.8 | 74.9 | 93.9 | 86.9 |
| | ReAct+PLAY2PROMPT | 100.0 | 75.7 | 99.6 | 87.5 |

sees way less gains, even when performance on the demonstration set improves. This suggests that 1) there is still quite a generalization gap between optimized demonstrations and the testing distribution, suggesting room for improvement; and 2) meta-information conveys a certain degree of information that may be easily picked up by larger models compared to smaller models. The information gained through tool play for these tools are still very helpful for the smaller model, which essentially bridges that information gap with PLAY2PROMPT.

**Optimization Model** $\mathcal{M}$    We further explore the effects of using a stronger optimization model $\mathcal{M}$, as they not only may generalize better, but also provide better self-reflection capabilities to enable better search. Due to computational constraints, we explore using LLaMA-3-70B in place of LLaMA-3-8B on the subset I3-Inst, and report results in table 4. We observe a fairly large improvement, essentially doubling the performance gain on this small subset, which suggest potential overall performance gains.

Table 4: Different $\mathcal{M}$ on I3-Inst. LLaMA-3-70B is used as $\mathcal{B}$ in this experiment.

| Method | $\mathcal{M}$ | I3-Inst |
|---|---|---|
| ReAct | - | 76.5 |
| ReAct+PLAY2PROMPT | LM-3-8B | 80.3 |
| ReAct+PLAY2PROMPT | LM-3-70B | 85.8 |

**Qualitative Analysis**    To illustrate how PLAY2PROMPT leverages tool play errors to optimize demonstrations and descriptions, we show a qualitative example in figure 2, where the tool documentation is outdated, specifying `start_date` and `end_date` instead of `from` and `to`, in addition to setting them to be required parameters when in fact they are optional. We show one query out of the three we generate for the demonstration set. In this case, PLAY2PROMPT in the early states is confused by contradictory information from the tool error and the documentation, but adds more detailed information and solves some of the queries that did not require the start and end dates. It ultimately starts exploring and ends up finding the correct parameter names, leading to superior performance. An additional example of a more typical improvement by PLAY2PROMPT is shown in appendix B.

## 4    RELATED WORK

**LLMs for Tool Use**    Recent years have witnessed significant advances in employing large language models (LLMs) as agents to master tool use for solving complex tasks (Mialon et al., 2023; Qin et al., 2024a), thereby enhancing LLMs' capabilities in areas such as multi-modal understanding (Gupta & Kembhavi, 2023; Surís et al., 2023; Wu et al., 2023), programming tools (Gao et al., 2023; Paranjape et al., 2023; Team et al., 2023; Zhang et al., 2023b; Cai et al., 2024), and other domain-specific functionalities. The conventional strategy involves training base models with tool-use data (Thoppilan et al., 2022; Dubey et al., 2024) or fine-tuning LLMs (Patil et al., 2023; Schick et al., 2023; Yang et al., 2023) to learn to use tools, which works well on specific tasks with a small fixed number of tools. Specifically, Parisi et al. (2022) explored tool play in a self-training context, aiming to automatically fine-tune a language model on a small number of tools. However, these approaches require continual learning as new tools are added, making the training process not scalable. Hao et al. (2023) addressed this by training tool embeddings that can be augmented onto fixed

Figure 2: An example of PLAY2PROMPT facing incorrect documentation. The beam search trajectory with the highest evaluation solve rate on the demonstration set is shown. At each state transition, a new description is explored based on error feedback.

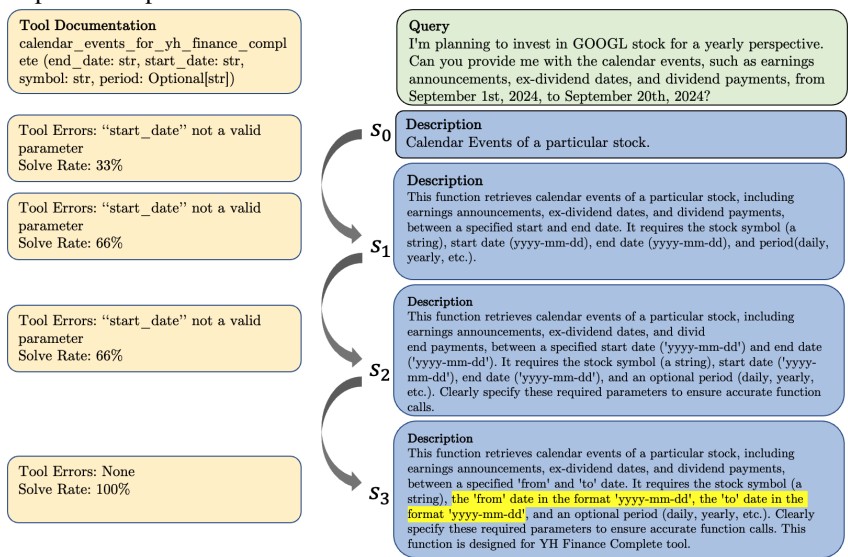

LLMs for plug-and-play usage; however, they still require labeled data to obtain the embeddings. Alternatively, LLMs can access tools via handcrafted meta-prompts or by being trained with tool-use instructions, and then supplied the tools during inference through prompts (Lu et al., 2023; Shen et al., 2023; Song et al., 2023; Qin et al., 2024b; Zhuang et al., 2024). With the increasing number of applications and tools in which LLMs are utilized, enhancing LLMs' tool-use capabilities for novel tools remains an important problem, which we explore and improve upon with PLAY2PROMPT.

**Tool Use Instructions and Optimization**   Tool documentation and example demonstrations are crucial components in prompting LLMs for effective tool use, as demonstrated by various studies. Hsieh et al. (2023) reported that documentation is more important than demonstrations for some tasks, and that LLMs often hallucinate tools when lacking proper documentation. Xu et al. (2023) investigated the effects of in-context example demonstrations on tool use techniques, observing diminished performance when such examples were omitted. To automate the generation of tool-use instances, Shen et al. (2024) proposed sampling tool calls from a graph of tool relations and back-instructing to construct queries, which relies on the availability of external tool graphs. In an effort to improve tool documentation, Yuan et al. (2024) utilized direct prompting to summarize and rewrite tool descriptions, but their approach relies on related documentation examples and lacks the ability to systematically search and optimize. While automatic prompt tuning methods (Pryzant et al., 2023; Wang et al., 2024) have been developed to adapt LLMs to domain-specific tasks by rewriting prompts, they typically depend on held-out testing sets to measure optimization quality, making them unsuitable for zero-shot tool instruction rewriting (Wu et al., 2024). These challenges highlight the necessity for approaches that can automatically optimize tool instructions and demonstrations without requiring labeled data or manual effort, which PLAY2PROMPT addresses by leveraging interactions with the tool itself.

## 5   CONCLUSION

We present PLAY2PROMPT, an automated framework that iteratively refines tool documentation and generates example tool usage demonstrations, enhancing the ability of large language models to utilize tools effectively in zero-shot settings. By employing a search-based trial-and-error approach with self-reflection, PLAY2PROMPT enables models to interact with tools, explore their function-alities, and improve both tool descriptions and demonstrations without the need for labeled data or extensive manual effort. This approach addresses the limitations of existing methods that rely on

handcrafted prompts or labeled data, offering a scalable and task-agnostic solution applicable to a wide range of tools and domains. Our experiments on StableToolBench demonstrated significant improvements over baseline methods for both open-source and closed-source models. By systematically enhancing the tool-use capabilities of LLMs, this work contributes to the development of AI agents that can autonomously adapt to new tools and challenges, extending their utility in real-world applications.

## LIMITATIONS AND FUTURE WORK

In this work, we generate proxy testing sets based only on a single given tool and do not cover multi-tool use. Scaling from single-tool scenarios to multiple tools can likely enhance LLM's tool use effectiveness. Additionally, for example demonstrations, we use rejection sampling to generate tool invocations first, which do not work for functions whose parameter space is too large, for instance parameters that take long ID string inputs or authentication tokens that require calls to other tools beforehand. Exploring multi-tool dependencies could potentially resolve this issue and improve tool play. In our work we focus on tool descriptions and demonstrations only, relegating other information as meta-information, which could be potential next steps to explore.

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

## A  Inference and Evaluation Details for StableToolBench

For GPT models, we keep the exact same setting as used in the original benchmark, except for requiring it to return an action from the provided toolset by supplying a flag to the OpenAI API. For LLaMA models, we adapt the ReAct prompts into its format but keep everything else fixed as much as possible. To provide fairer comparison, since LLaMA models do not have built in ways to restrict its tool calling to the provided tool set as GPT models can, we re-sample outputs for a certain amount of times if a tool hallucination falls outside of the tool set. This may also be quite easily done by constraining output tokens with certain syntax or grammar.

During inference, for PLAY2PROMPT we set the number of example demonstrations to 1, sampling temperature to 0.2 for LLaMA models.

## B  More Qualitative Examples

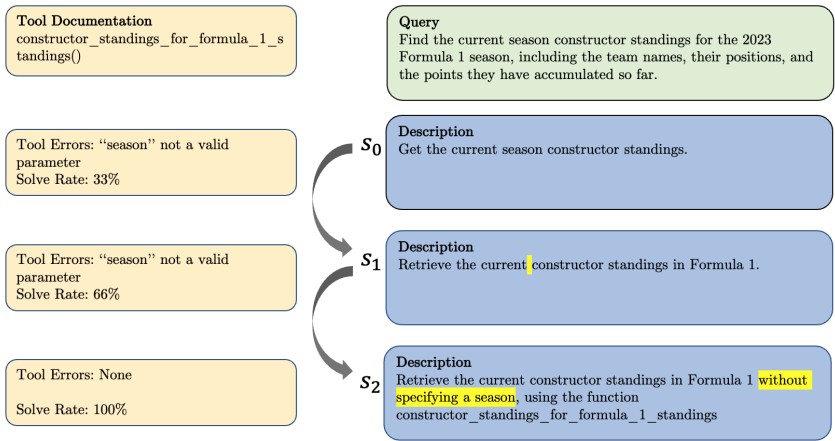

Figure 3: A typical example of PLAY2PROMPT assisting LLMs in correcting errors in generating parameter values. The base LLM gets confused by the query specifying the year, which PLAY2PROMPT first attempts to remove "year" from the description, and further explicitly prompts the base LLM to not use the parameter. The base LLM in this instance is LLaMA-3-8B.

