# OpenReview forum: "PLAY2PROMPT: Zero-shot Tool Instruction Optimization for LLM Agents via Tool Play"
_ICLR.cc/2025/Conference — ICLR 2025 Conference Withdrawn Submission_

### Official Review · Reviewer_6tG1 · 2024-11-02

**Soundness:** 2
**Presentation:** 2
**Contribution:** 3
**Rating:** 3
**Confidence:** 3

**Summary:**

The paper introduces PLAY2PROMPT, an automated framework that enhances the ability of large language models (LLMs) to utilize tools effectively in zero-shot settings.  The framework iteratively refines tool documentation and generates example tool usage demonstrations, allowing LLMs to explore tool functionalities without relying on external examples. PLAY2PROMPT employs a search-based trial-and-error approach augmented with self-reflection, enabling interaction with tools and iterative improvements to tool descriptions and demonstrations. The paper demonstrates PLAY2PROMPT's effectiveness through extensive experiments on real-world tasks, showing significant improvements in zero-shot tool performance across open- and closed-source models.

**Strengths:**

1. PLAY2PROMPT tackles the problem of optimizing tool usage in LLMs without relying on labeled data, which is a novel approach in the field of natural language processing and AI tool integration.
2. By focusing on zero-shot learning, PLAY2PROMPT extends the application of LLMs to new domains where labeled data may be scarce or non-existent, which is a significant contribution to the adaptability of LLMs.

**Weaknesses:**

1. There is a concern that the baseline(ReAct) and results(PLAY2PROMPT) in the experiments might be too weak, particularly given that the current SOTA on the [StableToolBench benchmark is around 70%](https://github.com/THUNLP-MT/StableToolBench?tab=readme-ov-file#model-experiments-results). For GPT-3.5-Turbo-1106, the result on StableToolBench is 62.2±0.8. I refer to the results from StableToolBench github repo. If this is an unreasonable comparison, please explain in detail.
2. There are 2 examples in Figure 2 and Figure 3 about single-tool scenarios, please provide some examples for multi-tool scenarios.

**Questions:**

1. PLAY2PROMPT used a score reward `r_t` to evaluate the quality of each state, during optimization. Please provide a visualization or quantitative analysis of how the reward r_t changes over iterations

---

### Official Review · Reviewer_m9z7 · 2024-11-02

**Soundness:** 3
**Presentation:** 3
**Contribution:** 2
**Rating:** 5
**Confidence:** 3

**Summary:**

This paper presents PLAY2PROMPT, a novel zero-shot tool instruction optimization framework designed for large language model (LLM) agents to improve tool use. The approach emphasizes "tool play," where the LLM explores the input-output behavior of tools iteratively, thereby generating optimized descriptions and usage examples without labeled data. PLAY2PROMPT refines tool documentation and validates tool usage by enabling LLMs to "play" with tools to learn effective usage patterns. The authors evaluate this framework on the StableToolBench benchmark, showing significant performance improvements for both open- and closed-source LLMs.

**Strengths:**

+ The approach of having LLMs “play” with tools to learn how to use them is a fresh take on tool optimization in a zero-shot setting. It’s creative, especially since it doesn’t require labeled data, just exploration and self-reflection.

+ The paper backs up its claims with extensive testing and solid methodology. The authors use structured search techniques and break down the process into distinct steps for refining tool documentation and generating example usage, which keeps it organized and effective.

+ The paper is generally easy to follow and well-structured. It clearly explains the framework, breaking down each component, so it’s straightforward to see how PLAY2PROMPT works.

**Weaknesses:**

- The method works well for single-tool scenarios, but it doesn’t yet support multi-tool use, which limits it in cases where tasks require switching between tools or combining multiple tools.

- Beam search, while effective, could be computationally demanding, particularly for larger models. Testing alternative search strategies might improve efficiency without sacrificing performance.

- The framework’s benefits vary with model size, and larger models seem to gain more from PLAY2PROMPT. Smaller models might not generalize as well, suggesting they may need more tailored adjustments.

- The main metric used, solvable pass rate, gives a basic view of improvement but doesn’t dive into specific types of failure cases, like whether certain tasks consistently remain unsolved.

**Questions:**

How much does PLAY2PROMPT rely on the base model’s initial understanding of tools? Would smaller, less powerful models need extra tweaks to achieve similar improvements?

---

### Official Review · Reviewer_5g5W · 2024-11-04

**Soundness:** 3
**Presentation:** 2
**Contribution:** 2
**Rating:** 5
**Confidence:** 4

**Summary:**

This paper addresses the critical issue of low-quality tool documentation when LLMs utilize external tools, which can significantly impact model performance. The authors propose Play2Prompt that employs beam search to generate multiple candidate documentations and demonstrations, selecting optimal ones through evaluation. Empirical results on StableToolBench demonstrate that the proposed method enhances the performance of various models, including Llama-3-8b, Llama-3-70b, and GPT-3.5-turbo.

**Strengths:**

1. The paper tackles a fundamental challenge in tool utilization of LLMs: the quality of tool documentation, which is crucial for effective model-tool interaction.
2. The proposed method demonstrates remarkable effectiveness and robustness, achieving significant improvements using Llama-3-8b rather than requiring more computationally intensive models like GPT-4, highlighting its practical applicability.
3. The comprehensive analysis of the impact of tool documentation and demonstrations provides valuable insights for future research directions.

**Weaknesses:**

1. The reward computation relies on LLM-based evaluation by prompting, raising concerns about the reliability of the search process. This dependency on LLMs for evaluation may introduce biases or inconsistencies in the quality assessment of generated documentations as LLMs may generate inaccurate scores.
2. The experimental evaluation would be more convincing with validation across additional datasets beyond StableToolBench to demonstrate broader applicability (e.g., BFCL[1], ToolQA[2], etc).
3. In the Ablation on Search Strategies section, it claims that the MC variant does not explore the search space well enough to reach better states. Given the widespread success of Monte Carlo Tree Search (MCTS) in LLMs[3,4], a comparative analysis between MCTS and the proposed beam search approach would provide valuable insights into their relative effectiveness for this specific application.
4. The paper would benefit from a clearer presentation, specifically through the inclusion of a high-level flowchart or algorithmic pseudocode before Methodology Section to enhance reader comprehension.

[1] Berkeley function calling leaderboard. https://gorilla.cs.berkeley.edu/blogs/8_berkeley_function_calling_leaderboard.html

[2] Yuchen Zhuang, et al., ToolQA: A Dataset for LLM Question Answering with External Tools. https://arxiv.org/abs/2306.13304

[3] Shibo Hao, et al., Reasoning with Language Model is Planning with World Model. http://arxiv.org/abs/2305.14992

[4] Weimin Xiong, wt al., Watch Every Step! LLM Agent Learning via Iterative Step-Level Process Refinement. http://arxiv.org/abs/2406.11176

**Questions:**

See the weaknesses part.

---

### Official Review · Reviewer_xXT2 · 2024-11-04

**Soundness:** 3
**Presentation:** 3
**Contribution:** 2
**Rating:** 3
**Confidence:** 3

**Summary:**

The manuscript describes a technique dubbed *play2prompt* which consists of two algorithmic proposals of improving the baseline use of tools by LLMs. The particular algorithm changes are complementing the baseline ReAct, a very well known and established algorithm for multi-step reasoning, including multi-step tool use. At a high-level, the proposed changes help the LLM make better use of a tool. This is done by using additional compute (e.g. MC sampling, BeamSearch sampling, Self-reflection) to identify description problems (e.g. tool parameters, general description of how to use the tool, etc) and construct some demonstrations instead of using the tool as zero-shot.

The results are mainly reported on StableToolBench, a recently introduced benchmark with many API call examples. Seemingly the benchmarks provides a way to work around offline service and use an API simulator. The models studied involve LLaMA-3-8B, LLaMA-3-70B and GPT-3.5. The authors report improvements for every models and category of the benchmark.

**Strengths:**

The main strengths of the manuscript consists of
* Addressing a real-world challenging problem, i.e. tools have a lot of variability in documentation, argumentation description and the models tend to struggle with using them outside well-studied domains.
* Leveraging a benchmark with many API calls, thus reflecting fairly well the problem discussed. Showcasing improvements on 2 model classes, one open-source (Llama) while and another one closed-source (GPT-3.5). At least two sizes as well for OSS (8B, 70B)
* Rather clean description of the algorithms involve in refining and improving the the tool use.

**Weaknesses:**

I think there is insufficient novelty in this work. To ground my feedback, I suggest these angles for discussion or improvements:
* The choice of baseline challenges the innovation brought by the manuscript. In practice, ReAct is very well-known at this point, so is the fact that zero-shot performance for Tool-use is not ideal. The authors do describe related work in auto-prompting, but do not make use of any method to create few-shot examples for tools to compare against. Besides the ones from the authors in Related Work, some other to call out are ART [1], DSPy [2].

* The benchmark is rather recently introduces and unfortunately, in my opinion, it dilutes the manuscript's claimed contribution. In particular, it is difficult to state whether the type of problems illustrated in Figure 2 (e.g. tools description with wrong arguments) are more frequently encountered in this benchmark. To concretely call out some others, could be Mint [3], APIBench [4]

[1] https://arxiv.org/abs/2303.09014
[2] https://github.com/stanfordnlp/dspy
[3] https://arxiv.org/abs/2309.10691
[4] https://arxiv.org/abs/2305.15334

**Questions:**

Looking forward to discuss weakenesses.

---

### Official Review · Reviewer_WpAJ · 2024-11-10

**Soundness:** 2
**Presentation:** 3
**Contribution:** 2
**Rating:** 5
**Confidence:** 4

**Summary:**

The authors acknowledge the challenge of zero-shot tool use for LLMs given the noisy nature of LLM documentation. The authors introduce, "PLAY2PROMPT" which is a system for LLMs to explore tool-use settings in simulation enabling them to refine the tool descriptions and examples. With Play2Prompt the authors are able to demonstrate 3-6\% points improvements for open-source (LLAMA) and closed-source (GPT-3.5) LLMs on StableToolBench.

**Strengths:**

1. The paper presents a comprehensive and complete technique to address what is often not focused upon - improving documentation for tools such that the LLM knows then to invoke them. This is fundamental to improve the LLM's ability calls tools in a zero-shot setting.

2. The technique of first sampling the tool invocation first (followed by rejection sampling), and then generating the corresponding query x and answer y, is a smart technique to introduce variance in the query distribution.

**Weaknesses:**

1. From the definition of the tool, v = (w, y, i) which includes the question, answer, and the invocation, the authors do not include the "environment" that the tool "invocation" is part of. While I understand the benefits of isolating the tool call from the environment, I wonder if by not capturing the information at all - would the system be loosing information relevant to adjudicate if the tool call is correct or incorrect?

2. In line 213, for generating the query "x", I'm wondering if the team had an ablation for 3-shot in-context prompt based query generation would be sufficient - or in other words, what's the benefit from the seemingly compute intensive technique of self-reflection and refinement?

3. How would the system handle multi-step or multi-turn tool calls? From line 285, the authors mention "an API service represents a tool that contains multiple sub-tools, with each sub-tool corresponding to f in our definition." which I would read as ~ creating single-step tool-call proxy for multiple tool calls? Which isn't true multi-step / single-step call?

**Questions:**

1. I'm curious on how using the definition in Section (2), would nested tool-call or a sequence of tool-call be supported? For example, if f = (u, I, g), would a nested tool call be g_1(g_o) in which case what would the description (u) be? A combination of the two calls, or a totally new call?
2. StableToolBench is a comprehensive benchmark and it is promising to see improvement performance. Does this generalize well to other zero-shot benchmarks as well, such as the Berkeley Function Calling Leaderboard?
3. Line 172: "Specifically, given a state st, an action at is generated by sampling from an optimization model M,
conditioned on the input-output information obtained from tool interactions." Why would the action "at" be determined by the optimization Model (M) and not a deterministic system?

---

### Note · Authors · 2024-11-25

I have read and agree with the venue's withdrawal policy on behalf of myself and my co-authors.